# Comparative analysis of three studies measuring fluorescence from engineered bacterial genetic constructs

**Jacob Beal** [1]*, **Geoff S. Baldwin**[2]*, **Natalie G. Farny**[3]*, **Markus Gershater**[4]*, **Traci Haddock-Angelli**[5]*, **Russell Buckley-Taylor**[2], **Ari Dwijayanti**[2], **Daisuke Kiga**[6], **Meagan Lizarazo**[5], **John Marken**[7], **Kim de Mora**[5], **Randy Rettberg**[5], **Vishal Sanchania**[4], **Vinoo Selvarajah**[5], **Abigail Sison**[5], **Marko Storch**[8], **Christopher T. Workman** [9], **the iGEM Interlab Study Contributors**[¶]

**1** Raytheon BBN Technologies, Cambridge, MA, United States of America, **2** Department of Life Sciences and IC-Centre for Synthetic Biology, Imperial College London, London, United Kingdom, **3** Department of Biology and Biotechnology, Worcester Polytechnic Institute, Worcester, MA, United States of America, **4** Synthace, London, United Kingdom, **5** iGEM Foundation, Cambridge, MA, United States of America, **6** Faculty of Science and Engineering, School of Advanced Science and Engineering, Waseda University, Tokyo, Japan, **7** Department of Bioengineering, California Institute of Technology, Pasadena, CA, United States of America, **8** London Biofoundry, Imperial College London, London, United Kingdom, **9** DTU-Bioengineering, Technical University of Denmark, Kongens Lyngby, Denmark

¶ Membership of iGEM Interlab Study Contributors is provided in S1 File Consortium Author List.
* jakebeal@ieee.org (JB); g.baldwin@imperial.ac.uk (GSB); nfarny@wpi.edu (NGF); m.gershater@synthace.com (MG); traci@igem.org (THA)

**Data Availability Statement:** All relevant data are within the manuscript and its Supporting information files.

## Abstract

Reproducibility is a key challenge of synthetic biology, but the foundation of reproducibility is only as solid as the reference materials it is built upon. Here we focus on the reproducibility of fluorescence measurements from bacteria transformed with engineered genetic constructs. This comparative analysis comprises three large interlaboratory studies using flow cytometry and plate readers, identical genetic constructs, and compatible unit calibration protocols. Across all three studies, we find similarly high precision in the calibrants used for plate readers. We also find that fluorescence measurements agree closely across the flow cytometry results and two years of plate reader results, with an average standard deviation of 1.52-fold, while the third year of plate reader results are consistently shifted by more than an order of magnitude, with an average shift of 28.9-fold. Analyzing possible sources of error indicates this shift is due to incorrect preparation of the fluorescein calibrant. These findings suggest that measuring fluorescence from engineered constructs is highly reproducible, but also that there is a critical need for access to quality controlled fluorescent calibrants for plate readers.

## Introduction

Effective characterization and reproducibility are amongst the core challenges for synthetic biology [1–4], and fluorescence measurement is still one of the key tools for quantifying cell

**Funding:** Partial support for this work was provided by NSF Expeditions in Computing Program 469 Award #1522074 as part of the Living Computing Project. Funder URL: https://www.nsf.gov/ The funders had no role in study design, data collection and analysis, decision to publish, or preparation of the manuscript. The following authors are employed by for-profit companies: Jacob Beal is employed by Raytheon BBN Technologies; Markus Gershater and Vishal Sanchania are employed by Synthace. These companies provided support in the form of salaries for these authors, but did not have any additional role in the study design, data collection and analysis, decision to publish, or preparation of the manuscript. The specific roles of these authors are articulated in the 'author contributions' section."

**Competing interests:** The authors of this manuscript have read the journal's policy and have the following competing interests: The authors received no specific commercial funding for this work. The following authors are employed by for-profit companies: Jacob Beal is employed by Raytheon BBN Technologies; Markus Gershater and Vishal Sanchania are employed by Synthace, and their work on this paper was thus indirectly supported by their salaries. This does not alter the authors' adherence to PLOS ONE policies on sharing data and materials.

behavior. Prior studies have shown that relative fluorescence can be reproduced within approximately 2-fold accuracy [5, 6]. More recently, large-scale interlaboratory studies conducted by the International Genetically Engineered Machine (iGEM) competition have demonstrated that such results can be improved and mapped to consistent and biologically relevant units with an appropriate choice of calibrants for fluorescence [7] and optical density (OD) [8]. Moreover, unit calibration protocols can provide better detection of biological protocol errors [7], as well as internal consistency checks against errors in the calibration process itself [8].

Reproducibility, however, can only be ensured by metrological traceability (defined by NIST as "establishment of an unbroken chain of calibrations to specified reference standards") and quality control of the reference materials used for calibration. We hypothesize that such a problem lies behind the order-of-magnitude difference in plate reader fluorescence values reported for equivalent constructs between the iGEM 2016 [7] and 2018 [8] interlaboratory studies, particularly since the flow cytometry fluorescence values for these two studies are close to one another. To investigate this hypothesis, we re-analyze all of the data from the 2016 interlaboratory study using the improved analytical methods of the 2018 study, as well as comparing with previously unpublished data from the iGEM 2017 interlaboratory study.

Across all three studies, we find similarly high precision in the calibrants used for plate readers (precision of the flow cytometry calibrants has already been established by NIST [9–11]). We also find that fluorescence measurements agree closely across the flow cytometry results and the plate reader results from 2017 and 2018, with an average geometric standard deviation of only 1.52-fold across the four sets of data. The 2016 plate reader results, however, are consistently shifted upward by an average of 28.9-fold. Analysis of possible sources of error indicates this shift is likely to have come from incorrect preparation of the fluorescein calibrant. These findings suggest that measuring fluorescence from engineered constructs is highly reproducible, but also that there is a critical need for access to quality controlled fluorescent calibrants for plate readers.

## Results

Interlaboratory studies were organized as part of the 2016, 2017, and 2018 International Genetically Engineered Machine (iGEM) competitions. Although not identical, the protocols used in the three studies are expected to be equivalent, and thus directly comparable. The studies all included components for examining the calibration protocol for plate readers and the fluorescence of cells transformed with GFP-expressing genetic constructs (flow cytometry calibrant precision having been previously established [9–11]). Here we compare the precision observed in the plate reader calibration measurements across the three years, then compare the experiments' measurements.

### Experimental data collection

Across three consecutive years, iGEM teams were invited to participate in an interlaboratory study on reproducibility of fluorescence measurements from engineered bacteria. In all three years, teams were provided with a set of calibration materials and a collection of engineered genetic constructs expressing GFP constitutively at a variety of levels. The 2016 study was designed to evaluate the reproducibility of fluorescence measurements with independent calibrants and was reported in [7]. In this study, teams were provided with five genetic constructs (illustrated in Fig 1, with complete details provided in Materials and Methods and S2 File DNA Constructs): a negative and positive control and three test constructs that were identical except for the promoter. The test constructs use promoters from the Anderson library [12]

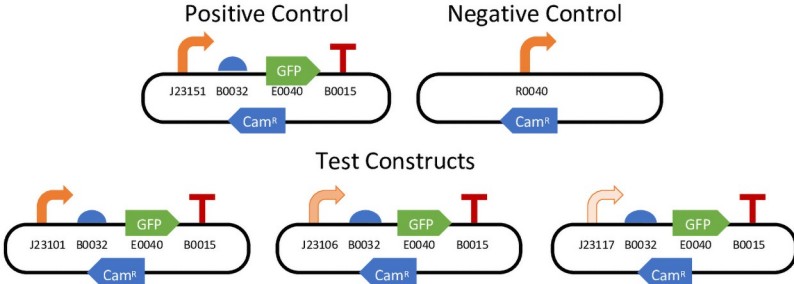

**Fig 1. Constitutive fluorescence genetic constructs measured in all three of the 2016, 2017, and 2018 iGEM Interlab Studies, diagrammed using standard SBOL visual symbols [13].**

expected to provide high, medium, or low expression levels. The 2017 and 2018 studies included the same constructs, plus three other constructs that were not shared between years and thus not analyzed here.

In each year, participants transformed *E. coli* and cultured two biological replicates for each construct, then measured these samples and calibrant materials. Results were input into a provided Excel calculation sheet. Data quality control was applied via a set of acceptance criteria detailed in S4 File Data Acceptance Criteria. Certain details varied, as the calibrant and protocols were modified in subsequent years in order to reduce variability and error (full details provided in S3 File Protocols). The mathematics behind the calibration protocols, however, predicts that they should all produce the same units. Thus, a key hypothesis of this comparative analysis is that the observed experimental values should agree across years.

In the 2016 study, as reported in [7], teams were asked to use *E. coli* K-12 DH5-alpha or TOP10 strains if available, with 63% using DH5-alpha, 23% using TOP10 and the remaining 14% using a variety of other strains. Teams were asked to measure with a plate reader (or spectrophotometer if no plate reader was available), and/or a flow cytometer. For unit calibration, teams were given moderate density colloidal silica (LUDOX HS-30) and 50 $\mu$M FITC for plate readers, and used SpheroTech calibration beads for flow cytometry. Plate reader OD was calibrated by comparing quadruplicate samples of LUDOX and water to the OD measured in a reference spectrophotometer. Fluorescence was calibrated with a serial dilution of FITC with PBS in quadruplicate.

The 2017 study, not previously published, was designed to reproduce the 2016 study and improve the reliability of the materials. All teams were required to use the DH5-alpha strain and measure using a plate reader, plus optional flow cytometry. The 50 $\mu$M FITC was changed to 50 $\mu$M unmodified fluorescein, improving solubility and simplifying preparation. The colloidal silica was changed to the denser LUDOX HS-40, improving precision by better separating OD calibration measurements. Since some teams' LUDOX had previously been frozen in shipping, kits were also provided with a freeze indicator to avoid using damaged calibrants.

The 2018 study, as reported in [8], was designed to improve OD calibration and put plate reader and flow cytometer data into equivalent units. As in 2017, all teams were required to use the DH5-alpha strain and measure using a plate reader, plus optional flow cytometry. The fluoroscein was reduced from 50 $\mu$M to 10 $\mu$M to better match the range of cellular fluorescence. The colloidal silica was changed to LUDOX CL-X, which is both denser and resistant to freezing. Teams were also provided with an additional OD calibrant and protocols, which are not included in this comparative analysis.

In 2016, participating teams produced 65 bulk fluorescence data sets and 14 flow cytometry data sets, but after retroactively applying the stricter acceptance criteria from 2017 and 2018

(S4 File Data Acceptance Criteria), only 36 of the bulk fluorescence sets are retained for the analysis presented here. In 2017, participating teams produced 188 accepted plate reader data sets and 3 flow cytometry data sets. The flow cytometry data sets are excluded from this analysis due to their small number. In 2018, participating teams produced 244 accepted plate reader data sets and 17 flow cytometry sets.

Critically, note that the differences between the protocols across the three years are not expected to have a significant effect on the values obtained for fluorescence expression. In particular:

- DH5-alpha and TOP10 are fairly similar strains, per strain records at the Coli Genetic Stock Center. All data sets from 2016 that passed the selection criteria were from DH5-alpha except for four from TOP10 and three from other strains.

- As the LUDOX has a relatively low OD, it should be in the linear range of the particle density to OD relationship [14], and the changes between the three different LUDOX variants should be accounted for by changes in the reference value provided.

- FITC and fluorescein are based on the same chemical chromophore with near identical chemical structure (http://sigmaaldrich.com) and fluorescence spectra (http://fluorophores.org/). It should be noted that FITC contains the chemically reactive isothiocyanate group for amine labelling and secondary affects associated with chemical reactivity cannot be excluded for this compound.

- Measured fluorescence is proportional to fluorescein concentration. This has been validated in prior reports [7, 8].

- Any differences in noise distributions or protocol variability should be observable as differences in the distributions of calibrant and cell measurements.

As such, the results of these three studies are subject to direct comparison to assess reproducibility, despite minor differences between the years. Moreover, successful reproduction will provide evidence in favor of the adaptability of the calibration protocols, separating them from dependence on any particular calibrant material.

### Robustness of plate reader calibration

We assess the robustness of the plate reader calibration protocols across the three years using the same two metrics as in [8]: replicate precision and residuals. For replicate precision, we compute the coefficient of variation (i.e., ratio of standard deviation to mean) for each set of technical replicates in each of the two calibration protocols. The more precise the execution of a protocol, the less variation there should be between technical replicates and the smaller this value should be. In [8], we found a high degree of precision for both the LUDOX/water OD calibration and the fluorescein dilution fluorescence calibration protocols. This proves to be true with similar distributions for the 2016 and 2017 studies as well, as shown in Fig 2.

For LUDOX/water, in 2016 the CV is $\leq 0.1$ for 84.6% and 87.7% of LUDOX and water replicate sets respectively; while in 2017 it is slightly worse at 79.8% and 83.5%. In 2018, it is slightly better at 86.9% and 88.1%. The fluorescein replicate sets are similar: in 2016 the CV is $\leq 0.1$ for 72.22% of replicate sets, while in 2017 it is 84.6% and in 2018 it is 76.9%. In short, though the specific numbers vary somewhat, in all three studies the vast majority of all calibrant replicate sets exhibit a high degree of precision across replicates.

For the fluorescein dilution series, we can also consider the residuals when the protocol is fit against the pipetting model from [8] (details provided in Materials and methods). With this

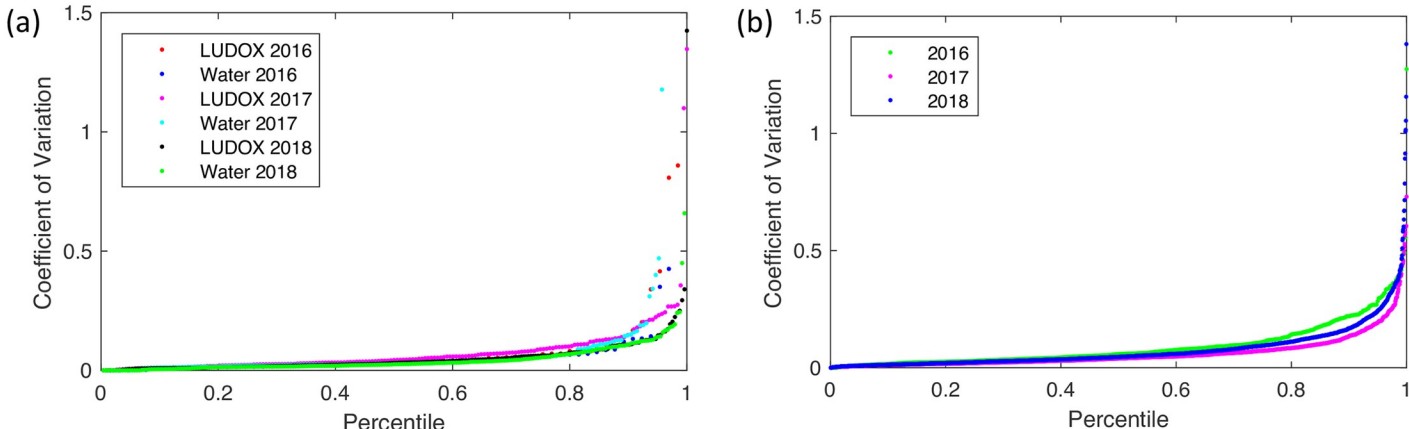

**Fig 2. Distribution of the coefficient of variation.** Distribution for valid replicate sets in LUDOX and water calibrants for OD (a) and fluorescein calibrant for fluorescence (b). All distributions show an overall high degree of precision and similarity, except for the small fraction of outliers in the LUDOX and water calibrants.

model, we are able to take into account not just one set of replicates but the entire series in order to assess potential issues in protocol execution, compensating for systematic pipetting error and identifying larger inconsistencies that cannot be corrected. To assess precision via model residuals, we first fit the pipetting model to each data set, then compare the predicted values to the experimental values; the closer the residual to one, the better the model fits the data and thus the more precise the protocol. Note that the LUDOX/water protocol cannot be assessed for residuals, since the dimensionality of its model matches the number of measurements, which is one of the reasons the particle dilution introduced in [8] is a preferable calibration protocol. That protocol, however, was developed for the 2018 study and not previously available.

Consistent with the results in [8], we find that the residual values are generally small and highly similar across the three years, as shown in Fig 3. For 2016, we find that 96.9% of all fluorescein residuals were within a range of $\leq 1.2$-fold (i.e., $0.8\bar{3} \leq$ residual $\leq 1.2$), while for 2017 this is true for 97.9% of residuals and for 2018 it is true for 98.0% of residuals.

In conclusion, both calibration protocols appear to operate with reproducibly high precision, as demonstrated by their similarity across all three studies in both coefficient of variation and residuals. Importantly, however, recall that the accuracy of these calibrations is not guaranteed by their precision.

## Fluorescence levels and accuracy

We now consider the focal question of biological reproducibility, comparing the fluorescence measurements from transformed cells. Here, the preferred unit is Molecules of Equivalent Fluorescein (MEFL) per cell, a close proxy for actual molecule count that can be used with both flow cytometry and plate reader data. In the 2018 study reported in [8], a microsphere particle calibrant for OD measurement allowed estimation of the number of cells (or equivalent obscuration) in each sample, which enables plate reader measurements to be converted to the same MEFL/cell units as flow cytometers, allowing direct comparison of flow cytometry and plate reader measurements. The 2016 and 2017 studies lacked the particle calibrant, however, and used only the LUDOX/water protocol for OD measurement calibration, producing units of MEFL/OD for plate readers. We thus first compare plate reader measurements in MEFL/OD units, then re-analyze the 2018 calibration data to produce a protocol-independent

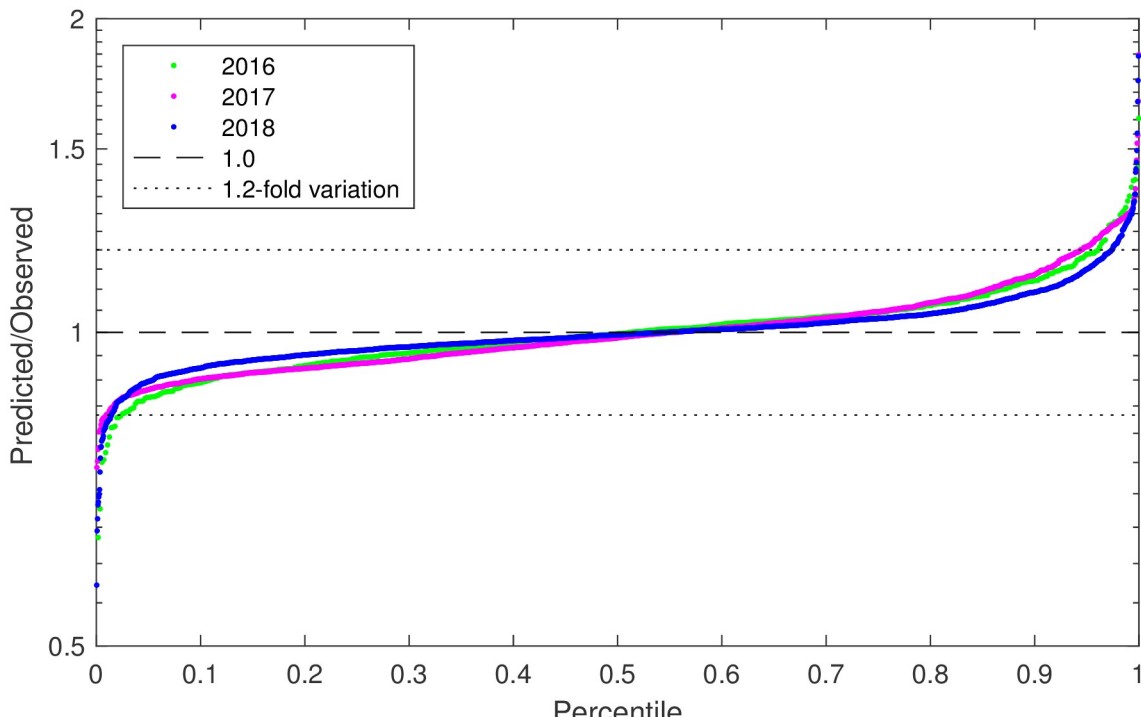

**Fig 3. Distribution of residuals for fluorescein calibrants.** Model fit residual distribution for every set of fluorescein replicates included in calibration computations for the 2016, 2017, and 2018 studies. Distributions for all three studies are similar, all indicating a high degree of precision.

particle/OD conversion factor that can be used to compare all three studies' plate reader data with flow cytometry data in MEFL/cell units.

Fig 4(a) shows the MEFL/OD fluorescence values computed from plate reader data for each of the three test constructs shared between all studies, excluding data with poor calibrant quality, control values, or colony growth (for details see Materials and methods on criteria for valid *E. coli* data). All three genetic constructs produce fairly tight distributions across all three years: the geometric mean of the geometric standard deviations across all nine conditions is only 2.04-fold. Moreover, the variability is consistently highest for the J23101 construct, which is known from [8] to have significantly higher variability related to problems in culturing. The geometric means across years of geometric standard deviations per genetic construct are 2.72-fold for J23101, 1.67-fold for J23106, and 1.89-fold for J23117. The actual values determined for the constructs, however, are not the same: while the 2017 and 2018 values match closely, with a geometric mean ratio of 1.51-fold, the 2016 values are consistently much higher, by a geometric mean ratio of 29.1-fold.

From [8], we know that the 2018 flow cytometry and 2018 plate reader measurements match well when compared in MEFL/cell units. The plate reader data in 2016 and 2017 studies cannot be directly converted to MEFL/cell units from the calibrants included in those studies. Since both the LUDOX/water calibrant and particle calibrant operate in the linear range of OD measurements, however, there should be a protocol-independent particle/OD ratio that can be applied to convert MEFL/OD units into MEFL/cell units. We compute this ratio by taking the ratio of the valid particles/Abs600 and OD/Abs600 conversion factors that were computed for each team in the 2018 study, finding this value to have a geometric mean across

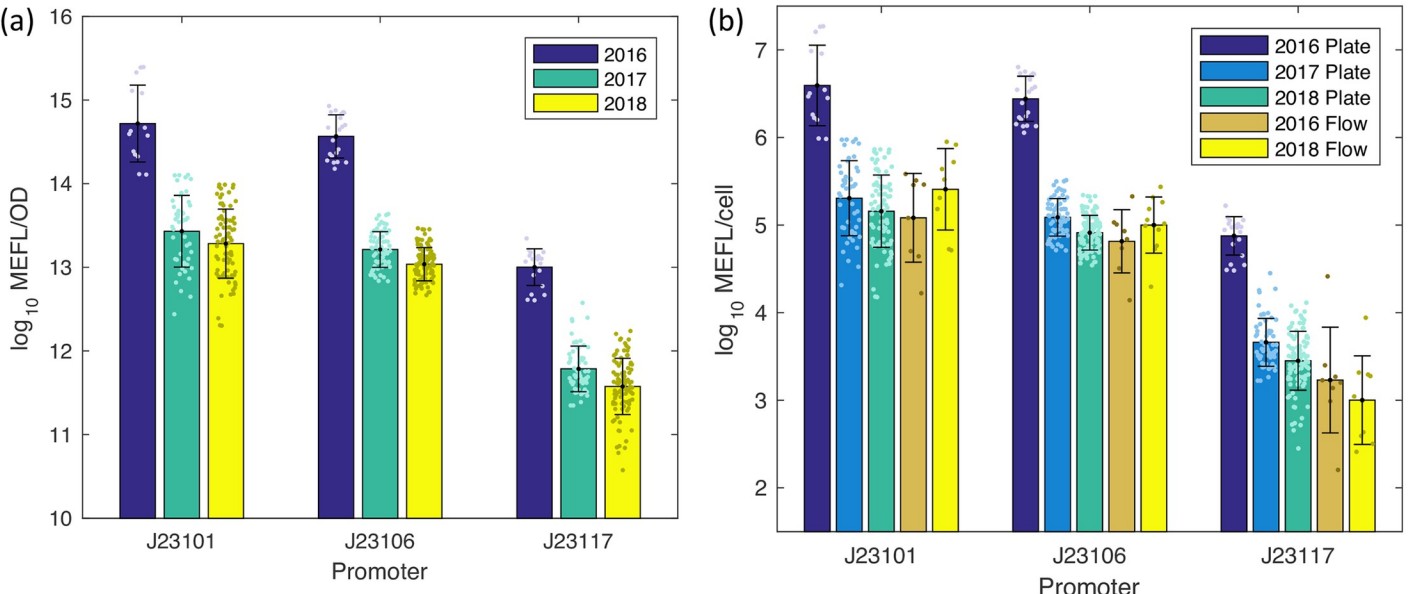

**Fig 4. Comparison of measured fluorescence of test genetic constructs.** Comparison of plate reader data in MEFL/OD units (a), and both plate reader and flow cytometry data in MEFL/cell units (b). Bars show geometric mean and standard deviation; dots show values from individual teams. Team count per condition is provided in S7 File Teams Per Condition.

teams of 1.33$e$8 particles/OD, with a highly consistent geometric standard deviation of only 1.59-fold.

Fig 4(b) compares the values obtained by applying this conversion factor to transform plate reader data into MEFL/cell values with the single-cell values directly obtained using calibrated flow cytometry in the 2016 and 2018 studies. Plate reader and flow cytometry are thus two independent modalities for obtaining the same units. Here, we see that the flow cytometry values match well between the two years, with a geometric mean ratio across genetic constructs of 1.77-fold. The flow cytometry values also match the 2017 and 2018 plate reader values: the geometric mean of the geometric standard deviations across all four of these year/modality conditions is only 1.52-fold, with the highest difference in the construct with the J23117 promoter. This difference is expected due to flow cytometers' greater ability to differentiate weak fluorescence levels (see discussion in [7, 8]).

The 2016 plate reader data, on the other hand, clearly does not match any of the other values, with a geometric mean ratio across genetic constructs of 28.9-fold higher than the geometric mean of values from the other year/modality conditions. The 2016 plate reader values are also much less biologically plausible, given that it estimates J23101 and J23106 at a mean 3.92$e$6 and 2.76$e$6 MEFL/cell respectively, yet *E. coli* has been estimated to have only 3–4 million total proteins per cell [15].

To diagnose the likely source of this inaccuracy, Fig 5 presents an end-to-end block diagram of the workflow used in all three studies, highlighting the opportunities for correlation between errors at each stage. At the end of the workflow, the same analysis code is applied to all of the data sets. Any bugs in the code would thus affect all of the data sets, so the analysis code is also unlikely to be the source of the inaccuracy. The stages executed at many sites— sourcing of cells and reagents, execution of the protocol, measurement of samples, and reporting of data—should be independent in the set of errors that occur. Many erroneous data sets are indeed detected and excluded, and the distribution of values for each year/modality

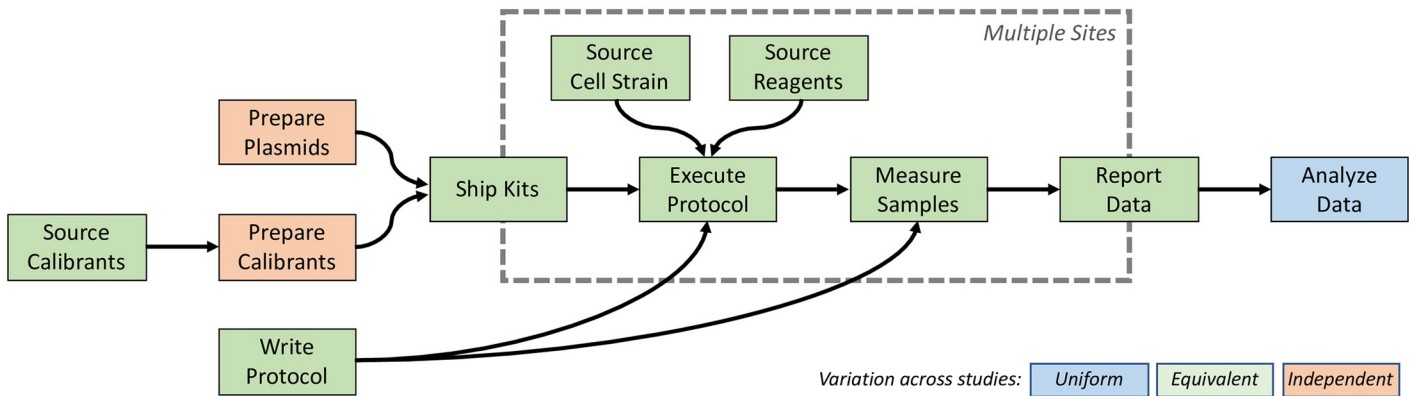

**Fig 5. Potential sources of error in studies.** Workflow block diagram of the different stages of each study, highlighting likely dependence or independence of potential errors in execution.

combination is much smaller than the inaccuracy of the 2016 plate reader data, indicating that this portion of the protocol appears highly reproducible and unlikely to be the source of inaccuracy. That narrows down the potential sources of inaccuracy to be in the materials provided to the teams.

If the issue was due to the plasmids, then it would require independent fabrication errors in all three plasmids, as well as that those independent fabrication errors should all produce a consistent ratio of increase in expression level. This combination is possible, but seems highly unlikely. An error in the protocol provided to the teams could create correlated errors in execution. The 2016 protocol did initially include an incorrect description of the fluorescein calibrant concentration as being 5 times higher than its intended value, but that error is unlikely to affect the results. First, the value was not used for execution of the protocol, only for interpreting the results, and the values used in our calculations are the raw measurements, not the interpreted values affected by the error. Second, the issue was detected early and the vast majority of teams did not begin the study until after the protocol was corrected. Another possibility is that the intended differences in the protocol might have caused some variation (e.g., the switch from FITC to fluorescein as fluorescence calibrant), but the magnitude of any of these differences is expected to be far smaller than the observed shift. Moreover, any issue that affected cells (as opposed to calibration) would have showed up in the corresponding flow cytometry data.

That leaves only the preparation of the plate reader calibrants themselves. As the fluorescent calibrant was prepared in a single batch for each year, an unforeseen reagent problem, such as reactivity associated with the isothiocyanate of FITC or its solubility in resuspension, a bad batch of reagent, or an error in fluorescein preparation could cause the full magnitude of the observed error. The commercial reagents involved, however, are common, stable, and subject to routine quality control. All together, this indicates that the most likely source of the observed inaccuracy in the 2016 plate reader data is an issue in the preparation of the fluorescein calibrant for that year, before the calibrant was packaged into kits and shipped to teams.

We further validated this conclusion with a post-study re-execution of fluorescence calibration series with unused 2016, 2017, and 2018 fluorescein calibrant samples. This re-execution yielded a 1.08-fold standard deviation between the 2017 and 2018 samples and mean 7.1-fold difference between these and the 2016 fluorescein samples (S2 Fig Replicate Calibrant Measurement). By this measure, it appears that error in preparation of the fluorescein calibrant accounts for approximately two-thirds of the log-scale difference between the 2016 plate reader

data and other year/modality conditions, leaving only a 4.1-fold difference remaining. Approximately half of this remainder can be absorbed by the standard deviation between other conditions, and the rest likely stems from some combination of the other potential issues discussed above. We speculate that the remaining difference may be related to the low OD of the LUDOX HS-30 used in 2016, which would amplify the effect of any inaccuracy in the measurement of its reference value.

## Discussion

The close relationship of four sets of values across two measurement modalities, three years, and varied protocols is evidence for both the accuracy and reproducibility of fluorescence measurements from cells transformed with synthetic constructs. Well-calibrated and controlled measurements may be expected to have no more than a two-fold geometric standard deviation. This finding depends on and supports the underlying calibration model, suggesting that it should also be able to adapt to different fluorescent reporters, cell species and cell morphologies, culture conditions, or instrument modalities such as luminescence or microscopy. Studies that fail to provide this degree of reproducible accuracy may suffer from issues in experimental protocol design and execution rather than any inherent variability of biology.

However, as the inaccuracy of the 2016 plate reader data shows, there is a pressing need for metrologically traceable fluorescence calibrants. Preparation of "home brew" calibrants is an inherently fragile endeavor, subject to the errors observed here unless there is a readily available reference to assure correctness. The situation for OD calibrants is better, as microspheres like those used in the 2018 study are already on the market, but the certified tolerances typically provided could be improved. Given the widespread use of plate readers, there is both a scientific need for improved metrology and a commercial opportunity for providing traceable calibrant materials.

Until such materials are available, it will be important to validate the accuracy of plate reader measurements by comparison with a second modality of measurement (e.g., flow cytometry) or with reference constructs (such as those reported in this study). Such methods, however, are inferior to traceable calibrants because some biological phenomena are expected to have different effects on the values reported by different instruments. For example, adherent cells have different OD than cells suspended in media, due to their inhomogeneous packing on the light path [14], but this spatial effect would not impact measurements in a flow cytometer after trypsinization. When flow cytometry is used as a substitute for traceable calibrants, this may cause confusion and mis-interpretation, while the availability of traceable calibrants would transform the situation into an opportunity to quantify adherence based on distortion. Many other such potential problems abound, which could be transformed into opportunities for quantification with the availability of traceable calibrants.

In summation, based on this comparative analysis, we make the following three recommendations:

- Investigators should calibrate all plate reader and flow cytometry measurements to reproducible units.

  - Fluorescence should be reported in units of molecules or Molecules of Equivalent calibrant (e.g., MEFL for a fluorescein calibrant)

  - OD should be reported in units of cells or Equivalent Particle Count (EPC).

- Journals and reviewers should require both plate reader and flow cytometry data to be reported in calibrated units.

- Commercial suppliers should develop traceable calibrant products.

Together, implementation of these recommendations will support a transformation in measurement culture and corresponding increase in reproducibility and reliability of results across a wide range of biological scientific and engineering applications.

## Materials and methods

### Calibration materials

Each participating team was given a sample of colloidal silica for calibration of OD and a sample of fluorescein for calibration of fluorescence. These materials varied slightly from year to year. In particular, the provided colloidal silica was:

- 2016: LUDOX HS-30 (Sigma-Aldrich #420824): one tube with 30% colloidal silica suspended in 1mL of water. Standard cuvette measurement in a spectrophotometer gave a reference OD value of 0.022.

- 2017: LUDOX HS-40 (Sigma-Aldrich #420816): one tube with 40% colloidal silica suspended in 1mL of water. Standard cuvette measurement in a spectrophotometer gave a reference OD value of 0.043.

- 2018: LUDOX CL-X (Sigma-Aldrich #420891): one tube with 45% colloidal silica suspended in 1mL of a water, with added ethylene glycol for anti-freeze purposes. Standard cuvette measurement in a spectrophotometer gave a reference OD value of 0.063.

The fluorescein standard was:

- In 2016, FITC standard tubes were prepared by combining 165.6 mg of FITC (Sigma-Aldrich F4274) powder with 0.1 L Dimethylformamide (DMF) to produce a 4.253 mM solution. Each tube received 11.76 uL of this solution (5.00e-8 M FITC), which was then vacuum dried for shipping. Resuspension in 1 mL PBS produces a solution with initial concentration of 50 $\mu$M FITC.

- In 2017, fluorescein standard tubes were prepared with 5.00e-8 moles fluorescein (Sigma-Aldrich #46970) in solution in each tube, which was then vacuum dried for shipping. Resuspension in 1 mL PBS produces a solution with initial concentration of 50 $\mu$M fluorescein.

- In 2018, fluorescein standard tubes were prepared with 1.00e-8 moles fluorescein (Sigma-Aldrich #46970) in solution in each tube, which was then vacuum dried for shipping. Resuspension in 1 mL PBS produces a solution with initial concentration of 10 $\mu$M fluorescein.

Each team providing flow cytometry data obtained their own sample of SpheroTech RCP-30–5A Rainbow Calibration Particles (SpheroTech). This calibrant is a mixture of particles with eight levels of fluorescence, which should appear as up to eight peaks (typically some are lost to saturation at the high end or instrument noise at the low end of the range). In 2016, all teams were assumed to use beads in the range of Lot AA01 to Lot AC01, for which manufacturer-supplied values are identical. If this assumption was violated, calibration values would be expected to shift by 1% to 8%, depending on lot. In 2018, teams reported the lot number to allow selection of the appropriate manufacturer-supplied quantification for each peak.

## Constructs, culturing, and measurement protocols

The genetic constructs supplied to each team for transformation are provided in S2 File DNA Constructs. The protocols for plate readers and flow cytometers, exactly as supplied, are provided in S3 File Protocols.

Note that in 2017 and 2018, additional constructs were provided to teams and are part of the protocol, but these are not analyzed in this manuscript because they were not repeated across years. Likewise, the 2018 protocol includes other means of calibrating OD measurements, which are not analyzed in this manuscript because they were not included in prior years and have already been presented in [8].

## Criteria for valid calibrant replicates

To analyze the precision of calibrants, the criteria from [8] were applied identically to all years to determine which replicate sets have sufficient data quality to be included into the analysis.

- **LUDOX/water**: A LUDOX/water calibration is considered valid if it fits the acceptance criteria in S4 File Data Acceptance Criteria, i.e., LUDOX has higher OD than water and water OD is not negative. For 2016, 62 of the 64 are valid, while for 2017 all 188 are valid and for 2018 all 244 are valid.

- **Fluorescein dilution**: A dilution level is defined to be *locally valid* if the measured value does not appear to be saturated either high or low. High saturation is determined by lack of sufficient slope from the prior level, here set to be at least 1.5x. Low saturation is determined by indistinguishability from the blank (i.e., zero fluorescein) replicates, here set to be anything less than 2 standard deviations of the blank values above the mean blank value. The valid range of dilution levels is then taken to be the longest continuous sequence of locally valid dilution levels and the calibration set is considered valid overall if this range has at least 3 valid dilution levels.

  - In 2016, of the 65 data sets, all 65 are valid.

  - In 2017, of the 188 data sets, 185 are valid and 3 are not valid, one having too few valid dilution levels, the second having too little slope, and the third having invalid PBS data.

  - In 2018, of the 244 data sets, 243 are valid and 1 is not valid, having an inconsistent slope indicative of pipetting problems.
    S1 Fig Fluorescein Valid Dilutions shows that the vast majority of fluorescein dilution data sets are at least valid in the middle range, with less saturation issues in each succeeding year.

## Unit scaling factor computation

Unit scaling factors are computed using the same methods as presented in [8], substituting alternative reference values as appropriate.

**LUDOX/water.** The scaling factor $S_l$ relating standard OD to Abs600 is computed as follow:

$$S_l = \frac{R}{\mu(L) - \mu(W)} \tag{1}$$

where $R$ is the measured reference OD in a standard cuvette (for 2016: 0.022 for LUDOX HS-30, for 2017: 0.043 for LUDOX HS-40, and for 2018: 0.063 for LUDOX CL-X), $\mu(L)$ is the mean Abs600 for LUDOX samples, and $\mu(W)$ is the mean Abs600 for water samples.

No residuals can be computed for this fit, because there are two measurements and two degrees of freedom.

**Fluorescein dilution.** The scaling factor $S_f$ for relating molecules of fluorescein to arbitrary fluorescent units is computed as one parameter of a fit to a model of systematic pipetting error $S_p$.

If we ignore pipetting error, then the model for serial dilution has an initial population of calibrant $p_0$ that is diluted $n$ times by a factor of $\alpha$ at each dilution, such that the expected population of calibrant for the $i$th dilution level is:

$$p_i = p_0(1 - \alpha)\alpha^{i-1} \tag{2}$$

In the case of the specific protocols used here, $\alpha = 0.5$. For the fluorescein dilution protocol used in 2016 and 2017, $p_0 = 3.01e15$ molecules of fluorescein and in 2018, $p_0 = 6.02e14$ molecules of fluorescein.

The model for systematic pipetting error modifies the intended dilution factor $\alpha$ with the addition of an unknown bias $\beta$, such that the expected biased population $b_i$ for the $i$th dilution level is:

$$b_i = p_0(1 - \alpha - \beta)(\alpha + \beta)^{i-1} \tag{3}$$

We then simultaneously fit $\beta$ and the scaling factor $S_p$ to minimize the sum squared error over all valid dilution levels:

$$\epsilon = \sum_i |\log\left(\frac{b_i}{S_p \cdot (\mu(O_i) - \mu(B))}\right)|^2 \tag{4}$$

where $\epsilon$ is sum squared error of the fit.

The residuals for this fit are then the absolute ratio of fit-predicted to observed net mean $\frac{b_i/S_p}{\mu(O_i)-\mu(B)}$ for all valid levels.

**Application to *E. coli* data.** The Abs600 and fluorescence a.u. data from *E. coli* samples are converted into calibrated units by subtracting the mean blank media values for Abs600 and fluorescence a.u., then multiplying by the corresponding scaling factors for fluorescein and Abs600.

## Criteria for valid *E. coli* data

Data from each team were accepted to the study only if they met a set of minimal data quality criteria, including values being non-negative, the positive control being notably more fluorescent than the negative control, and measured values for calibrants decreasing as dilution increases. In 2017 and 2018, acceptance criteria were applied in advance, while for 2016 we apply the same criteria retrospectively, accepting 36 of 65 data sets. Full details are provided in S4 File Data Acceptance Criteria.

For analysis of *E. coli* culture measurements, a data set was only included if both its fluorescence calibration and OD calibration were above a certain quality threshold. The particular values used for the two calibration protocols were:

- **LUDOX/water**: Coefficient of variation for both LUDOX and water are less than 0.1.

- **Fluorescein dilution**: Systematic pipetting error has geometric mean absolute residual less than 1.1-fold.

Measurements of the cellular controls were further used to exclude data sets with apparent problems in their protocol: those with a mean positive control value more than 3-fold different than the median mean positive control.

Finally, individual samples without sufficient growth were removed, that being defined as all that were either less than the 25% of the 75th percentile Abs600 measurement in the sample set or less than 2 media blank standard deviations above the mean media blank in the sample set.

### Flow cytometry data processing

Flow cytometry data for 2016 was analyzed separately by each team, who then supplied the event count and geometric mean for each measured sample. Flow cytometry data for 2018 was processed using the TASBE Flow Analytics software package [16], using the recommended practices for gating, background subtraction, and bead-based calibration. Additional details and examples are provided in S5 File Flow Cytometry Data Processing for 2018.

### Statistics and reproducibility

As reproducibility is the main subject of this study, see the Results section above for its full presentation. In addition to the discussion of statistical analyses in the Results section, we note the following details of statistical analyses:

- Coefficient of variation (CV) is computed per its definition, as the ratio of the standard deviation to the mean.

- Fluorescence values are analyzed in terms of geometric mean and geometric standard deviation, rather than the more typical arithmetic statistics, due to the typical log-normal distribution of gene expression [17].

Data analysis was performed with Matlab.

## Supporting information

**S1 File. Consortium author list.** List of 2708 additional consortium authors comprising the iGEM Interlab Study Contributors.
(PDF)

**S2 File. DNA constructs.** SBOL 2 formatted file [18] containing DNA constructs for the 2016 iGEM Interlab Study.
(XML)

**S3 File. Protocols.** Protocols provided to teams for use in data collection for each year and instrument.
(PDF)

**S4 File. Data acceptance criteria.** Quality control criteria for inclusion of data for each team.
(PDF)

**S5 File. Flow cytometry data processing for 2018.** Details of how flow cytometry data processing was conducted for 2018 data.
(PDF)

**S6 File. Complete data.** JSON files containing of all input data sets, plus all results of analysis per Materials and methods above. For flow cytometry data, only the per-sample statistical

summary of each sample is included. In both cases, team names are omitted and data reported in an arbitrary order in order to anonymize data sets.
(ZIP)

**S7 File. Teams per condition.** Spreadsheet indicating the number of team datasets with valid data points included for each construct and calibration / measurement condition in Fig 4.
(XLSX)

**S1 Fig. Fluorescein valid dilutions.** Fraction of data sets with valid fluorescein dilution levels for each dilution number in the series.
(PDF)

**S2 Fig. Replicate calibrant measurement.** Replicate measurements of fluorescent calibrants.
(PDF)

## Acknowledgments

This document does not contain technology or technical data controlled under either the U.S. International Traffic in Arms Regulations or the U.S. Export Administration Regulations.

## Author Contributions

**Conceptualization:** Jacob Beal, Geoff S. Baldwin, Natalie G. Farny, Markus Gershater, Traci Haddock-Angelli, Meagan Lizarazo, Kim de Mora, Randy Rettberg.

**Data curation:** Jacob Beal, Natalie G. Farny, Traci Haddock-Angelli, Vinoo Selvarajah.

**Formal analysis:** Jacob Beal.

**Investigation:** Geoff S. Baldwin, Natalie G. Farny, Markus Gershater, Traci Haddock-Angelli, Russell Buckley-Taylor, Ari Dwijayanti, Daisuke Kiga, John Marken, Vishal Sanchania, Vinoo Selvarajah, Abigail Sison, Marko Storch, Christopher T. Workman.

**Methodology:** Jacob Beal, Geoff S. Baldwin, Natalie G. Farny, Markus Gershater, Traci Haddock-Angelli, Russell Buckley-Taylor, Ari Dwijayanti, Daisuke Kiga, John Marken, Vishal Sanchania, Vinoo Selvarajah, Abigail Sison, Marko Storch, Christopher T. Workman.

**Project administration:** Jacob Beal, Natalie G. Farny, Traci Haddock-Angelli.

**Writing – original draft:** Jacob Beal, Natalie G. Farny.

**Writing – review & editing:** Jacob Beal, Geoff S. Baldwin, Natalie G. Farny, Markus Gershater, Traci Haddock-Angelli, Russell Buckley-Taylor, Ari Dwijayanti, Daisuke Kiga, Meagan Lizarazo, John Marken, Kim de Mora, Randy Rettberg, Vishal Sanchania, Vinoo Selvarajah, Abigail Sison, Marko Storch, Christopher T. Workman.

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
