## [Decision Letter · Decision Letter 0]

12 Apr 2021

PONE-D-21-04191

Meta-Analysis of Three Studies Measuring Fluorescence from Engineered Bacterial Genetic Constructs

PLOS ONE

Dear Dr. Beal,

Thank you for submitting your manuscript to PLOS ONE. After careful consideration, we feel that it has merit but does not fully meet PLOS ONE’s publication criteria as it currently stands. Therefore, we invite you to submit a revised version of the manuscript that addresses the points raised during the review process.

We look forward to receiving your revised manuscript.

Kind regards,

Pasquale Avino, Ph.D.

Academic Editor

PLOS ONE

Journal Requirements:

2.  Thank you for stating the following in the Financial Disclosure section: "Partial support for this work was provided by NSF Expeditions in Computing Program 469 Award #1522074 as part of the Living Computing Project. Funder URL: https://www.nsf.gov/

And competing Interest: "The authors of this manuscript have read the journal’s policy and have the following competing interests: The authors received no specific commercial funding for this work. The following authors are employed by for-profit companies: Jacob Beal is employed by Raytheon BBN Technologies; Markus Gershater and Vishal Sanchania are employed by Synthace, and their work on this paper was thus indirectly supported by their salaries. This does not alter the authors’ adherence to PLOS ONE policies on sharing data and materials."

We note that one or more of the authors are employed by a commercial company: Raytheon BBN Technologies and Synthace.

4.  Please consider amending the title to more accurately reflect the nature of the work. We note that 'meta-analysis' typically refers to an analysis following a systematic review and utilising a specific framework.

5.   Please note that authors are responsible for ensuring that anyone named in the Acknowledgments agrees to be named (https://journals.plos.org/plosone/s/submission-guidelines#loc-acknowledgments).

Reviewers' comments:

Reviewer #1: In this manuscript, Beal et al conduct a metanalysis of fluorescence from bacterial strains. The data were generated by participants in the 2016, 2017, and 2018 IGEM competitions. The analysis conducted by the authors reveals high reproducibility in fluorescence across laboratories and years but also uncovers a large anomaly in fluorescence measurements from 2016. The authors conduct further analysis and suggest that the anomaly stems from errors in preparation of the fluorescent standard. The authors conduct an experiment that supports this

hypothesis. The authors conclude the article by providing recommendations for investigators working in synthetic biology. Overall, this is a well-written article, and the conclusions are generally supported by the data. By dealing with the often-overlooked issue of experimental reproducibility, the study represents an important contribution to the field of synthetic biology and should be accepted with only minor revisions.

We request that the authors consider addressing the following points:

Line 12: metrological traceability – unclear term – what does this mean to the reader not familiar with the field of instrument calibration? It would be useful to define this term here.

Line 98 – do you have a citation or reference for the statement that DH5-alpha and TOP10 are fairly similar? How do you define “fairly similar”?

Line 105 – do the authors have a reference for the statement that FITC and fluorescein are near-identical compounds with near-identical spectra.

Line 185 – “analysis of the 2018 study calibrant data finds this value to be 1.33e8” … how did the authors come up with this number?

Line 246-248 – can the authors speculate if there are particular factors that may contribute to the remaining variation not explained by issues in fluorescence calibration?

Reviewer #2: The manuscript by Beal and colleagues describes a meta-analysis of calibrated flow cytometry and plate reader data from the iGEM inter-lab study across 2016, 2017 and 2018. The main results are that calibration of both data types provides reproducibility across laboratories and machines. This message is very important for the synthetic biology field if it is really going to mature into a true engineering discipline. Another point from the paper is that errors in the calibration protocol can be disastrous for metrology and it is interesting to see the possible impact.

Overall, I think the authors have done a thorough job in collating the data across years and the message of the paper should (hopefully) have a big impact in the field. I have one minor comment:

When talking about the different protocols across years, the authors say the results across years are expected to be equivalent, and thus directly comparable. While this may be true for the expected values, it is not obvious to me that the noise distributions should be the same and changing concentrations etc will affect the variability. Can the authors add a comment on this?

---

## [Author Response · Author response to Decision Letter 0]

5 May 2021

See attached cover letter / response to reviewers

---

## [Editor Report · Decision Letter 1]

14 May 2021

Comparative Analysis of Three Studies Measuring Fluorescence from Engineered Bacterial Genetic Constructs

PONE-D-21-04191R1

Dear Dr. Beal,

We’re pleased to inform you that your manuscript has been judged scientifically suitable for publication and will be formally accepted for publication once it meets all outstanding technical requirements.

Kind regards,

Pasquale Avino, Ph.D.

Academic Editor

PLOS ONE

---

## [Editor Report · Acceptance letter]

27 May 2021

PONE-D-21-04191R1 

Comparative Analysis of Three Studies Measuring Fluorescence from Engineered Bacterial Genetic Constructs 

Dear Dr. Beal:

I'm pleased to inform you that your manuscript has been deemed suitable for publication in PLOS ONE. Congratulations! Your manuscript is now with our production department. 

Kind regards, 

on behalf of

Professor Pasquale Avino 

Academic Editor

PLOS ONE